# Prognostic Factors in Extremity Soft Tissue Sarcomas Treated with Radiotherapy: Systematic Review of the Literature

**DOI:** 10.3390/cancers15184486

**Published:** 2023-09-09

**Authors:** Arthur Lebas, Clara Le Fèvre, Waisse Waissi, Isabelle Chambrelant, David Brinkert, Georges Noël

**Affiliations:** 1Radiotherapy Department, ICANS, 17 Rue Albert Calmette, 67100 Strasbourg, France; a.lebas@icans.eu (A.L.); c.lefevre@icans.eu (C.L.F.); i.chambrelant@icans.eu (I.C.); 2Radiotherapy Department, Léon Bérard Center, 28 Rue Laennec, 69008 Lyon, France; waisse.waissi@lyon.unicancer.fr; 3Orthopedic Surgery Department, University Hospital of Hautepierre, 1 Rue Molière, 67200 Strasbourg, France; david.brinkert@chru-strasbourg.fr

**Keywords:** soft tissue sarcoma of the extremities, radiotherapy, prognostic factors, complications

## Abstract

**Simple Summary:**

Soft tissue sarcomas of the extremities are rare tumors with various prognostic factors. The gold standard curative treatment relies on surgery with negative margins. However, some prognostic factors of these tumors, or patients, may require additional RT treatment to improve oncological outcomes. We present a systematic review following the PRISMA guidelines, summarizing the pertinent literature on eSTS treated with RT. The great heterogeneity among these studies and the lack of statistical power have made it difficult to clearly identify significant prognostic factors that would benefit from this treatment. An analysis of these factors is proposed and provides valuable insights to optimize radiotherapy treatments for patients with eSTS. Based on our review, the acute and chronic serious adverse effects of multimodal treatment combining limb-sparing surgery and RT remain low, leading to favorable functional outcomes.

**Abstract:**

Soft tissue sarcomas of the extremities are rare tumors with various prognostic factors. Their management is debatable due to their inconsistent results within the literature and the lack of large prospective studies. The objective of this systematic review is to analyze the available scientific data on prognostic factors concerning the characteristics of the patients, the disease and the treatments performed, as well as their potential complications, on studies with a median follow-up of 5 years at minimum. A search of articles following the “PRISMA method” and using the PubMed search engine was conducted to select the most relevant studies. Twenty-five articles were selected, according to preestablished criteria. This review provides a better understanding of the prognosis and disease outcome of these tumors. Many factors were described comparing the frequency of occurrence according to the studies, which remain heterogeneous between them. Significant factors that could orient patients to radiotherapy were highlighted. These positive prognostic factors provide valuable insight to optimize radiotherapy treatments for patients treated for soft tissue sarcoma of the extremities.

## 1. Introduction

Soft tissue sarcomas are a heterogeneous group of tumors with more than 80 histologic subtypes arising in mesenchymal tissues [1]. It represents approximately 1% of all adult malignant diseases, and the extremities are the most common site of origin (60%), with a predisposition to the lower limbs [2,3,4]. The management and prognosis of these tumors vary depending on the anatomical location [5]. For localized soft tissue sarcomas of the extremities (eSTSs), the gold standard curative treatment relies on surgery with negative margins [6,7].

Previous studies demonstrated that local control (LC) and overall survival (OS) following limb-sparing surgery combined with preoperative or postoperative radiation therapy (RT) were similar to amputation but with improved functional and psychological outcomes [8,9,10,11]. This multimodal approach became the standard of care in eSTS with intermediate- and high-risk features. Perioperative RT is now recommended to improve the LC of patients with at least one of the following factors: positive margins, grade 2 and 3 tumors, deep-seated tumors and tumors ≥ 5 cm in size [6,7].

Although LC rates were similar between pre- and postoperative RT [12], treatment doses and irradiated volumes varied, resulting in different toxicities [13]. However, because of the rarity of these tumors, only a few prospective randomized studies are available [14]. Among these, a minority exclusively focused on sarcomas of the extremities [9,11]. In addition, most of the retrospective studies on this topic have a follow-up time limited to a few years, which may conceal some long-term benefits or toxicities [15,16,17].

As a result, some prognostic factors of LC, survival, distant metastasis, or complications differ according to the limited studies and remain poorly understood. Tumor grade has been shown to be a prognosticator and was retained in the American Joint Committee on Cancer (AJCC) staging system [18,19,20], but other studies have determined a wider range of prognostic factors, including patient and disease characteristics and treatment schedule or parameters [18,21,22].

To the best of our knowledge, there is currently no systematic review available on the potential benefits of RT in relation to prognostic factors for eSTS exclusively.

Thus, the purpose of the present work was to conduct a systematic review to clarify the importance of the prognostic factors in a subset of patients treated with eSTS, and to provide valuable insight to optimize radiotherapy treatments.

## 2. Materials and Methods

Article research followed the “PRISMA method” [23]. Articles corresponding to the terms ‘((radiotherapy) OR (radiation)) AND ((extremities) OR (extremity) OR (limb) OR (limbs)) AND ((sarcoma)) AND ((soft tissue))’ were searched in the PubMed database (https://www.ncbi.nlm.nih.gov/pubmed; accessed on 30 September 2022). Studies published between 1980 and 30 September 2022, written in English or French, were included. The selected studies reported patients treated for eSTS (except Ewing and trunk sarcoma) with RT. Only studies including at least 20 patients and a median follow-up of at least five years were included in the analysis. These articles mentioned the prescribed RT dose, LC or complications and reported prognostic factors.

## 3. Results

### 3.1. Identified Studies

Overall, 2099 articles were retrieved from the PubMed database. Among these, 2008 articles were excluded because they did not meet the inclusion criteria after abstract screening. Among the 91 remaining articles, 58 were excluded after reading the full text because of inclusions of the eSTS of the trunk, an average follow-up time for patients less than five years or a lack of data on radiation therapy doses and fractionation. One additional study was found through a hand search. Ultimately, 25 studies (24 retrospective studies and one randomized trial) were included in this current systematic review (Figure 1).

### 3.2. Patient Population

Twenty-five studies reported data on 3264 patients treated for eSTS with combined treatments, including limb-sparing surgery and RT. The data are reported in Table 1. The median range of the total sizes of the population varied from 23 to 412 patients. The M/F sex ratio was 1.18 (1588 males and 1337 females). The median age varied from 44 to 65 years old. The median follow-up was 6 years (ranging from 5 to 17.9 years). An extremity STS was defined as a tumor located between the shoulder and the distal part of the upper limb and between the buttock and the distal part of the lower limb. Some authors excluded pelvis and iliac fossa tumors for the lower limb. Sarcomas were located in the lower and upper limbs for 2430 lesions (78%) and 695 (22%), respectively. Two studies focused exclusively on the hand–wrist and foot–ankle regions [24,25].

The main pathologies were undifferentiated pleomorphic sarcoma (UPS), previously called malignant fibrous histiocytoma (MFH), synovial sarcoma and liposarcomas, including myxoid, round-cell and pleomorphic sarcomas. One study specifically studied liposarcoma [26], and another exclusively studied UPS [27]. Despite several different histological classification systems, most authors used the FNCLCC histological grading system, and others used UICC or AJCC TNM classification. The median tumor size (maximal diameter) varied from 3.2 to 14.2 cm. Some authors focused only on high-grade sarcoma [28] or included both intermediate- and high-grade sarcoma [29,30]. In all the selected studies, at least 2518 grade 2/3 sarcomas were studied, but three studies did not specify the grade of the sarcomas [31,32,33].

**Table 1 cancers-15-04486-t001:** Patient characteristics of the 25 studies selected for this review.

Series	Country	Sample Size	Age	Sex M:F	Location Type LL:UL	Median Follow-Up (Months)	More Frequent Histologic Subtype in Series	Tumor Size (cm)	Tumor Grade	Margin Status	Total Median Dose (Gy) ±Boost /Dose per Fraction
Cannon et al. [34], 2006	USA	412	49 (8–92)	1:1 (206–206)	LL:412	111.6 (14.4–372)	UPS 42% Liposarcoma 22% Synovial sarcoma 13%	8 (1.2–30) >5:304 <5:107	I:17 II:119 III:276	Positive/uncertain: 63 Negative:349	50 (44–70) 60 (50–72)
Folkert et al. [35], 2014	USA	319 (EBRT: 154; IMRT 165)	54 (17–89)	NA	2.9:1 (238–81)	60	UPS 37% Liposarcoma 28% Synovial sarcoma 9% Leiomyosarcoma 5%	<10:EBRT 84, IMRT 92 >10:EBRT 70, IMRT 73	High grade: EBRT 120; IMRT 143 Low grade: EBRT 34, IMRT 22	Positive/close margin No: EBRT 93, IMRT 8 Yes: EBRT 61, IMRT 85	50 (48–50.4) 63 (18–70.2)
Roeder et al. [36], 2018	Germany	259	55 (3–89)	1.7:1 (162–97)	4.2:1 (209–50)	54 (2–231)	Liposarcoma 31% UPS 27% Synovial sarcoma 15% Leiomyosarcoma 7%	Median: 8	High grade:236	R0:185 R1:74	45 (20–60.4) + 12 (7.5–20)/1.8–2
Alektiar et al. [28], 2002	USA	204	49 (16–89)	1:1 (103–101)	1.7:1 (128–76)	67	NA	3.2 <3:109 >3:88	High:204	NA	63 (30–70)/1.8–2 EBRT + BRT: 35–50 + 16–30 BRT:45
Goertz et al. [27], 2020	Germany	192	64.5 (18.3–89.9)	1.2:1 (106–86)	1.9:1 (126–66)	61.2	UPS 100%	≤5:69 >5:123	I:8 II:69 III:115	R0:179 R1:11 R2:2	60 (25–70)
McGee et al. [37], 2012	USA	173	57 (18–86)	1.2:1 (94–79)	1.9:1 (114–59)	124.8 (3.6–385.2)	UPS 51% Liposarcoma 18%	<5:8 >5–10:33 >10:13 Unknown:44	High grade: 154	Negative: 70% Marginal or microscopically positive: 30%	65 (49–74)/once or twice daily
Kneisl et al. [38], 2017	USA	162	≤50:61 >50:101	0.9:1 (78–84)	3.4:1 (125–37)	61.2 (9.6–243.6)	NA	≤5:56 >5:106	II–III:120 I:42	Positive:16 Close:26 Negative:117 Unknown:3	50 63
Beane et al. [39], 2014	USA	141 (71:No RT, 70:RT)	No RT:59.9 ± 2.2 RT:58.6 ± 3.2	1.2:1 (78–63)	3.1:1 (107–35)	214.8 (12–348)	NA	0–1.9:No RT 6, RT 5 2–4.9:No RT 19; RT 24 5–9.9:No RT 25, RT 27 >10:No RT 21, RT 13	I:No RT 19; RT 22 II:No RT 26; RT 24 III:No RT 21; RT 20	Positive (<1 mm): No RT11, RT 7 Negative; close (≤1 cm): No RT 20, RT 12 Negative; wide (>1 cm) No RT 5, RT 13 Negative; not specified:No RT 7, RT 11 R0:No RT 27, RT 27	45 + 18/1.8
Khanfir et al. [40], 2003	France	133	44 (16–88)	1.2:1 (73–60)	2.5:1 (92–37)	120 (36–300)	UPS 30% Synovial sarcoma 21%	6 (1–20)	I:36 II:55 III:36	R0:100%	50 (36–65)
Choong et al. [32], 2001	Australia	132	43.8 (10.1–83.9)	1.2:1 (71–61)	1.5:1 (79–53)	98.4 (18–210)	UPS 35% Liposarcoma 34% Fibrosarcoma 15% Leiomyosarcoma 7%	5 (0.7–30)	I:59 II:73	Marginal:39 Wide:91 Radical:2	62 (30–71) ±13.5 (3.6–20)
Felderhof et al. [41], 2013	Netherlands	118	NA	1:1 (58–60)	3.1:1 (89–29)	93 (9–192)	Myxoid liposarcoma 14% Leiomyosarcoma 13% Synovial sarcoma 12% UPS 6%	<5.0:46 5.1–10.0:43 >10.0:29	I–II:28 III:90	Involved:29 Marginal:75 Wide:12 Unknown:2	60/2 66/2
Dogan et al. [42], 2019	Turkey	114	44 (15–82)	1.1:1 (60–54)	2.6:1 (82–32)	60	UPS 26% Liposarcoma 25% Synovial sarcoma 13%, Fibrosarcoma 11%	7 (3–26) <5:41 5-<15:44 >15:29	I–II:13 III:101	Involved:25 Marginal:72 Wide:12 Unknown:5	60.9 (44–70)/1.8–2
Cheng et al. [43], 1996	USA	112	18–88	1.3:1 (63–49)	NA	63.6 (16–192)	UPS 45% Liposarcoma 21% Synovial sarcoma 12%	NA	NA	Intralesional:20 Marginal:26 Wide:66	48.2 ± 16.6
Mullen et al. [30], 2012	USA	96	49 (26–75)	1.2:1 (53–43)	5:1 (80–16)	111.3	Liposarcoma 26% UPS 22% Leiomyosarcoma 3%	14.2 (8–35)	II:25 III:23	R0:80 R1:15 R2:1	44 ± 16/2
Tanabe et al. [29], 1994	USA	95	52 (17–97)	1.8:1 (61–34)	6.7:1 (87–13)	66 (16–236)	UPS 43% Liposarcoma 24%, Synovial sarcoma 8%	0.1–5:16 5.1–10:27 10.1–15:33 15.1–20:9 >20:15	II:46 III:54	Positive:24 Negative:71	50 (38–70)/2
Blaes et al. [33], 2010	USA	89	50 (7–88)	1.4:1 (52–37)	LL: 89	87.6 (9.6–262)	NA	NA	NA	NA	63 (20–70.2)/1.8–2
Talbert et al. [24], 1990	USA	78	NA	1.4:1 (45–32)	1:1 (39–39)	94.8	Synovial sarcoma 32% UPS 11% Epithelioid sarcoma 9%	<2:16% 2–4.9:56% >5:28%	I–II:5 III:73	NA	62 (45–75)/2
Dickie et al. [31], 2009	Canada	74	58–63	0.8:1 (32–42)	LL:74	89	NA	NA	NA	NA	64 (57–71)
Wanebo et al. [44], 1995	USA	66	48 (17–77)	1.1:1 (34–32)	2.7:1 (48–18)	84	UPS 20% Synovial sarcoma 18.2% Liposarcoma 16.7%	<5:40% 5.1–10:25% 10.1–15:22% >15:13%	I:2 II:9 III:55	Wide:38 Radical:19 Amputation:4 Limited:2	30/3 35/3.5 46/2
Le Péchoux et al. [45], 1999	France	62	44 (15–76)	1.6:1 (38–24)	3.4:1 (48–14)	72	Synovial sarcoma 27% UPS 21% Liposarcoma 11% Neurosarcoma 10%	9.5 (1.0 21.0) >5:43	I:10% II:52% III:38%	Marginal:24 Incomplete:16	50 (45–65) ± 5–20/2 45/1.5 × 2
Dincbas et al. [46], 2014	Turkey	60	<50:35 ≥50:25	1.6:1 (37–23)	7.6:1 (53–7)	67 (8–268)	Synovial cell sarcoma 35% Liposarcoma 23% UPS 22% Leiomyosarcoma 7%	<12:23 ≥12:37	I:18 II:14 III:28	Marginal:31 Wide:24 Radical:5	35/3.5 46/2 50/2
Pao et al. [47], 1990	USA	50	52 (18–91)	1.5:1 (30–20)	1.8:1 (32–18)	70 (28–168)	Liposarcoma and UPS 60%	<5:22 5–10:18 >10:10	I:11 II:8 III:31	R0:10 R1:31 R2:8	60 (45–69)
Lee et al. [48], 2012	South Korea	43	NA	1.3:1 (24–19)	3.3:1 (33–10)	70 (5–302)	Liposarcoma 33% Synovial sarcoma 23% UPS 19%	7 (1.1–20)	I:11 II:13 III:19	Negative:20 Close (<2 cm):12 Positive:11	60 (50–74.4) /1.8–2
Issakov et al. [26], 2006	Israel	38	51.1 (18–84)	1.1 (20–18)	11.7:1 (35–3)	67 (9–123)	Liposarcoma 100%: - myxoid 55% - round cell 34% - pleomorphic 11%	NA	II–III:100%	Wide:10 Marginal:3 Involved:25 2nd attempt for marginal/involved: Wide 13, marginal 12, involved 3	63/1.8 70/1.8
Schoenfeld et al. [25], 2006	USA	23	64	0.8:1 (10–13)	0.9:1 (11–12)	132 (14.4–310)	UPS 39% Synovial sarcoma 17%, Dermatofibrosarcoma 9%, Leiomyosarcoma 9%	NA	High grade:18 Low grade:4 Undetermined: 1	Intralesional:0 Marginal:10 Wide:11 Radical:2	50.4 64.8 (60–74.4)/1.2 × 2

EBRT = external beam radiation therapy; F = female; Gy = Gray; IMRT = intensity-modulated radiation therapy; LL = lower limbs; M = male; Mo = months; NA = not available; RT = radiotherapy; UL = upper limbs; UPS = Undifferentiated pleomorphic sarcoma; y = year(s).

### 3.3. Treatments

#### 3.3.1. Schedule of Radiotherapy

Fourteen studies included patients treated with adjuvant RT only, three with neoadjuvant RT only, and eight studies included patients treated with both neoadjuvant and adjuvant RT. When specified, the interval between surgery and postoperative RT was between 3 and 6 weeks in most of the studies [24,28,34,37,45,47,48]. The interval between preoperative RT and surgery varied from 10 days to 6 weeks [29,30,34,44,46].

#### 3.3.2. Irradiation Technique

Most of the authors did not report the irradiation technique. Three-dimensional conformal radiotherapy (3D-CRT) was used for all patients in two studies [25,41]. McGee et al., Roeder et al., and Dincbas et al. used 3D-CRT after 1993, 1995, 2000, respectively [36,37,46]. One study used 2D-RT, 3D-CRT and intensity-modulated radiotherapy (IMRT) [38]. In total, 165 (51.7%) patients were treated with IMRT in the study by Folkert et al. [35]. Two studies used brachytherapy (BRT) [28,40]. Three studies [26,33,47] used external beam radiation therapy (EBRT) with photons only, four [24,29,36,41] combined photons and electrons, four used cobalt 60 and photons [25,42,46,48] and one did not specify [34].

#### 3.3.3. Set Up

Only a few authors described their treatment setup and planning. McGee et al. explained that the target volumes were defined by fluoroscopic simulation until 1993 when computed-tomography-based three-dimensional (3D) images became available [37]. Computed tomography was also used at least by other authors [25,35,38,41,46]. To improve the reproducibility of the setup, individual immobilization devices are often used [28,40,41]. If MRI was available, matching was reported in two studies [37,46].

#### 3.3.4. Radiation Therapy Prescription

All doses and fractionations are mentioned in Table 1. In most of these trials, conventional fractionation of RT was used, whereas hyperfractionated RT was used in three trials [25,37,45] and hypofractionated RT in one trial [39]. A prescription isodose was specified only by Roeder et al. at 90% [36]. Only Dincbas et al. specified the biologic effective doses (BED) for hypofractionated and conventional treatment: 70 BEDGy_3.5Gy_, 47.3 BEDGy_10Gy_, and 72.3 BEDGy_3.5Gy_, 55.2 BEDGy_10Gy_, respectively [46].

Concerning volumes and fields, most of the authors specified that the margins were ≥5 cm longitudinally around the tumor bed for postoperative RT and between the gross tumor volume (GTV) and the clinical target volume (CTV) for preoperative RT. It reached 10 cm in two studies [26,49]. In the axial plane, margins ranged from 1 cm to 10 cm. Three studies specified that they adapted the prescribed volume to anatomic barriers [36,41,46]. Planning target volume (PTV) margins were 1 cm in two studies [41,46]. Only two authors specified the field dimensions; for Talbert et al., the median length of the field was 12 cm (range, 5–29 cm) [24], and for Blaes et al., the median was 31 cm (range, 16–58.5 cm) [33]. In two studies where patients were treated with EBRT in the groin, thigh, or knee, radiation, records specifically reviewed the proportion of circumference of the femur that received the entire prescribed radiation dose [33,34]. In the study of Cannon et al., the dose was delivered to the entire circumference of the femur in 160 (47%) patients or to none of or a partial circumference in 174 (52%) [34]. For Blaes et al., 49 patients (58%) had 100% of their femoral circumference within the radiation field, and 35 (39%) had a partial volume of femur in it [33]. For two other studies, the authors reviewed what percentage of the extremity (lower and upper limb) was treated [41,48]. Lee et al. concluded that the entire circumference of an extremity was never treated [48], and Felderhof et al. reported that the full circumference of the extremity was included for 12 patients (10%) [41]. Felderhof et al. also reported that a joint had been included in the radiation field for 73 patients (62%) [41].

A target boost was reported in 11 articles [24,28,30,32,36,37,39,42,43,45,46]. The boost dose ranged from 5 Gy to 20 Gy for EBRT, from 16 Gy to 30 Gy for BRT [28], and from 7.5 to 20 Gy for intraoperative electron radiation therapy (IOERT), prescribed to the 90% isodose [36]. Dogan et al. used boost margins of 2 cm around the tumor bed and the scar [42]. Only a few authors have described the indications for boost. For Mullen et al. and Alektiar et al., a boost was performed if the surgical margin was positive; for Roeder et al., an IOERT boost was performed for all patients [36]. In studies that used it, a boost with EBRT was delivered from 11% to 100% of patients depending on the studies. In studies with boost as BRT, a boost was used for six patients (11%) [28].

In the postoperative RT arm, Alektiar et al. used low-dose rate brachytherapy in 53 (60%) of 88 patients as an exclusive treatment for 47 patients with a dose of 45 Gy and a median dose rate of 0.41 Gy/h and as a boost for 6 patients with doses from 16 to 30 Gy [28]. In the study by Khanfir et al., 9 (15%) of 69 patients received postoperative BRT without precision regarding the dose [40].

#### 3.3.5. Chemotherapy

CT was used in 18 studies, adjuvant CT was the most commonly used in 15 studies [25,27,28,29,33,34,35,36,37,40,41,42,45,47,48] with doxorubicin-based CT, and neoadjuvant CT was used in 11 studies [24,26,27,29,30,36,41,44,45,46,48] and was composed mainly of protocols including doxorubicin or isolated limb perfusion with tumor necrosis factor alpha (TNF-α) and melphalan. Finally, CT was used concomitantly with RT in three articles [24,34,39]. Only a few authors have explained the indications of CT. According to the studies, they retained gross disease, large tumors, positive margins, or high grades, and patients were deemed to be at very high risk of recurrence [24,42,44].

### 3.4. Local Control

The overall results and prognostic factors are summarized in Table 2.

#### 3.4.1. Local Control with Only Preoperative Radiotherapy

The 5-year LC rates ranged from 81.0% to ≥98.5%. No data were available for 10 years. In the study by Tanabe et al., high-grade tumors had the worst LC (*p* = 0.05) [29]. Surgical margin status was a prognostic factor in univariate analysis, with a 5-year LC rate of 91% in margin-negative patients versus 62% in margin-positive patients (*p* = 0.005) [29], whereas it was not significant for Dincbas et al. [46]. Both authors did not demonstrate tumor size as a predictive factor of LC [29,46]. For Dincbas et al., in multivariate analysis, the LC rate was significantly better for patients irradiated with hypofractionation compared with conventional fractionation (*p* < 0.05) [46].

#### 3.4.2. Local Control with Only Postoperative Radiotherapy

The 5- and 10-year LC rates ranged from 75% to 100% and from 70.4% to 100%, respectively. Tumor depth, margin status, size, grade, adjuvant CT, re-excision, and sex were not found to be significant prognostic factors for LC [24,26,28,37,39,40,41,42,45,47,48]. Some factors have been found to worsen LC in some studies, such as age > 50/55 years, central tumor location (shoulder/groin), and UPS histological type [28,37,40]. However, these factors were disputed in another series [24,39,41,42].

A significant benefit of adjuvant RT was reported by Khanfir et al. with 5- and 10-year LC rates of 86% and 81% in the RT group and 69% and 61% in the no-RT group (*p* = 0.01) [40] and by Beane et al. with a significant reduction in local recurrence (LR), with 25% operated on alone and 1.4% in those who received adjuvant EBRT (*p* = 0.0001) [39]. However, Alektiar et al. did not find any significant benefit of adjuvant RT on LC, with either EBRT or BRT. Additionally, there was no difference in LC between BRT and EBRT [28].

Several characteristics of RT were studied. For Talbert et al., there was no LC rate difference according to the kind of radiation, photons, or electrons [24]. For Dogan et al., LC was significantly worse in patients who received less than 60 Gy (*p* = 0.03). Furthermore, a dose higher than 60 Gy for margin-positive patients led to a significantly better LC rate (*p* = 0.04) [42]. However, this factor was not retrieved by McGee et al. [37]. No significant difference was observed with a hyperfractionated schedule [45]. The addition of boosts was not studied in the articles included in this review.

#### 3.4.3. Local Control with Both Pre- and Postoperative Radiotherapy

Articles were composed of each of the two modalities, and some included preoperative RT plus postoperative boost [30,32,33]. The 5- and 10-year LC rates ranged from 67.6% to 92.4% and from 83.0% to 91.0%, respectively. While in the series of Goertz et al., LC was improved by adjuvant RT in multivariate analysis (HR: 0.41 [0.25–0.67); *p* < 0.001), no significant benefit was shown with neoadjuvant RT [27]. In contrast, no significant difference favoring either treatment schedule was found in three other studies [35,36,43]. Positive margins were found two times as a predictor of poor LC [27,36]. RT was mainly performed postoperatively in the studies of Goertz et al. and Roeder et al. (93.8% and 83%, respectively) [27,36]. RT significantly improved LC rates for patients with positive margins in the series of Choong et al., which used postoperative RT, preoperative RT or a combination of both (*p* < 0.001) [32]. Techniques of RT with IOERT or EBRT, regardless of the dose, did not influence LC in the series of Roeder et al. [36], but Folkert et al. showed that IMRT was a significant positive independent predictor of LR (HR: 0.458 [0.235–0.891; *p* = 0.02). This study was predominantly represented by postoperative RT (87%) [35].

#### 3.4.4. Chemotherapy and Local Control

In multivariate analysis, Dogan et al. demonstrated a 5-year LC benefit by adding postoperative CT with doxorubicin and ifosfamide to postoperative RT for patients with high-grade and large tumors (*p* = 0.03). However, they did not mention whether CT was administered concomitantly with RT [42]. Mullen et al. showed that patients receiving neoadjuvant CT concomitantly with RT with mesna, doxorubicin, ifosfamide, and dacarbazine (MAID) tended to have a nonsignificantly better prognosis than those who did not receive any neoadjuvant CT [30], and comparable results were not confirmed in other series [35,36].

In total, only a few prognostic factors for LC were identified in these studies. RT significantly increased LC in several articles, but the schedule seems to not impact LC, and the dose remains a source of debate. Similarly, other potential prognostic factors were largely disputable, indicating the heterogeneity of the series in terms of patients, tumors, treatments, and techniques.

### 3.5. Disease-Free Survival and Distant Control

The overall results and prognostic factors are summarized in Table 3.

The 5- and 10-year disease-free survival (DFS) rates ranged from 42% to 87% and from 30% to 87%, respectively. Kneisl et al. observed a 44% reduction in the risk of recurrence or death with RT (*p* = 0.069) [38]. Some authors found a significant improvement in DFS with preoperative RT [30], while others did not find any significance according to the timing of RT [43].

For Roeder et al., factors that retained significance in the prognosis of DFS in multivariate analysis were disease status (primary/recurrent), grade, margin status, and metastases prior to/at IOERT time [36], while Issakov et al. found that margin status was not significant [26].

A trend for better RFS in women was described by Dogan et al. (*p* = 0.07) [42]. IMRT did not significantly improve DFS, with a 5-year DFS of 57.2% (95% CI, 49–65.4) in the IMRT group and 56.3% (95% CI, 48.1–64.5) in the non-IMRT group. Several studied factors did not reach significance for DFS, RT delivery or not [28], fractionation of the RT [46] or kind of technique [35].

The 5- and 10-year distant control (DC) ranged from 46.0% to 95.2% and from 59.0% to 81.0%, respectively. Six articles described the median time to metastasis, which ranged from 14 months to 36 months, with a median time of 22 months [24,32,36,37,40,46]. The lung was the first metastatic site, reaching up to 94% [37]. Bone was the second metastatic site when specified [24,42,46]. Tumor size was significantly retrieved as a prognostic factor for DC in four studies [24,32,40,41], with different thresholds of 3 cm [24] or 5 cm [24,40]. Low grade was significantly correlated with a higher DC in three studies [32,36,40]. Liposarcomas were correlated with a higher DC than leiomyosarcoma [36]. For Mullen et al., DM-free survival was significantly improved by preoperative CT [30].

**Table 3 cancers-15-04486-t003:** Disease-free survival, distant control and predictive factors of patients with extremity soft tissue sarcomas treated with radiotherapy.

	**More Frequent Histologic Subtype in Series**	**Preoperative RT**		**Postoperative RT**		**Preoperative RT**		**Postoperative RT**	
		**5 y DFS**	**10 y DFS**	**5 y DFS**	**10 y DFS**	**5 y DC**	**10 y DC**	**5 y DC**	**10 y DC**
Wanebo et al. [44]	UPS 20%, Synovial sarcoma 18%, Liposarcoma 17%	44% (7 y)	NA			46% (7 y)	NA		
Dincbas et al. [46]	Synovial sarcoma 35%, liposarcoma 24% UPS 22%	48.1%	NA			51.8%	NA		
Talber et al. [24]	Synovial sarcoma 32%, UPS 11%, Epithelioid sarcoma 9%			61%	51%			83%	74%
Le Péchoux et al. [45]	Synovial sarcoma 27%, UPS 21%, Liposarcoma 11%			42% (30–54)	NA			NA	NA
Alektiar et al. [28]	NA			NA	NA			80% (74–86)	NA
Khanfir et al. [40]	UPS 30%, Synovial sarcoma 21%			NA	NA			71% (63–78)	59% (48–68)
Issakov et al. [26]	Liposarcomas 100%			NA	51%			NA	61%
Lee et al. [48]	Liposarcoma 33% Synovial sarcoma 23%, UPS 19%			67.9%	NA			73.3%	NA
McGee et al. [37]	UPS 51%, Liposarcoma 18%			NA	NA			82%	81%
Felderhof et al. [41]	Myxoid liposarcoma 14%, Leiomyosarcoma 13%, Synovial sarcoma 12%			64%	44%			69%	63%
Dogan et al. [42]	UPS 26%, Liposarcoma 25%, Synovial sarcoma 13%			60%	52%			NA	NA
		**Preoperative + Postoperative RT**							
		**5 y DFS**		**10 y DFS**		**5 y DC**		**10 y DC**	
Choong et al. [32]	UPS 35%, Liposarcoma 34%, Fibrosarcoma 15%	NA		NA		95.2% ± 2%		NA	
Schoenfeldet al. [25]	UPS 40%, Synovial sarcoma 17%, Neurofibrosarcoma 9%	87%		87%		NA		NA	
Cannon et al. [34]	UPS 42%, Liposarcoma 22%, Synovial sarcoma 13%	NA		62%		71%		67%	
Mullen et al. [30]	UPS 22%, Liposarcoma 16%	77% (MAID) vs. 42% (Control)		65% vs. 30%		80% vs. 48%		77% vs. 43%	
Folkert et al. [35]	UPS 37%, Liposarcoma 28%, Synovial sarcoma 9%	56.8% (51.4–62.8)		NA		NA		NA	
Roeder et al. [36]	Liposarcoma 31%, UPS 27%, Synovial sarcoma 15%	61%		58%		69%		66%	
Cheng et al. [43]	UPS 45%, Liposarcoma 21%, Synovial sarcoma 12%	Pré 56% ± 15%/Post 67 ± 12%		NA		NA		NA	

Black boxes mean ‘not applicable’; DC = distant control; DFS = disease-free survival; MAID = mesna, doxorubicin, ifosfamide, dacarbazine; NA = not available; RT = radiotherapy; y = year(s).

### 3.6. Overall Survival

The overall results and prognostic factors are summarized in Table 4.

For patients treated with preoperative RT, the 5-year OS rates ranged from 59.0% to 68.3%. No data were available for the 10-year OS. For those treated with postoperative RT, the 5-year OS varied from 62% to 80%, and the 10-year OS rates varied from 51% to 82%. Beane et al. reported a 20-year OS of 71% [39]. For studies including both preoperative RT and postoperative RT, the 5- and 10-year OS rates ranged from 56% to 96% and from 38% to 91%, respectively. RT significantly increased OS rates for Kneisl et al. and Goertz et al. when compared to no RT, with 5-year OS rates of 67.6% vs. 48.4%, respectively (*p* < 0.001) [27,38].

In the series of Tanabe et al., a trend of better OS rates was reported for patients who had local recurrences but who previously received RT compared to those who did not receive any RT [29]. Although the 5- and 10-year OS rates were higher in the group of patients who received RT than in those who did not, the difference did not reach significance in two studies [39,40].

Comparatively, similar conclusions were obtained by Cheng et al., even after stratification according to AJCC stages [43]. Different RT doses were studied (<60, 60–66 and >66 Gy [37], ≤56, 60, and 66 Gy [41], and <60 and ≥60 Gy) but did not impact OS [42]. Schedule [43], fractionation [37,45], or technique [35] also did not impact OS.

Local and distant recurrences were significantly associated with lower OS rates in four studies [36,37,40,41]. High grades and high stages seemed to be the most reported prognostic factors responsible for lower OS. High grade was a prognostic factor reported in six studies [27,29,36,40,44,48], and high stage was reported in three studies [43,44,47]. Positive margins were a significant negative factor of OS in three series [27,44,45] but failed to be retrieved in six series [26,29,37,40,41,47]. Metastatic disease led to a low median OS, ranging between 7 and 14 months [41,44]. In most articles, a trend was observed for better OS in women, without being significant, except for McGee et al. (*p* = 0.04) [37].

Adjuvant CT was not associated with a significant benefit in two studies [29,42], while neoadjuvant CT improved survival for Mullen et al. At 10 years, the OS for the MAID arm was 66% versus 38% for the control arm (*p* = 0.003) [30].

**Table 4 cancers-15-04486-t004:** Overall survival and predictive factors of patients with extremity soft tissue sarcomas treated with radiotherapy.

	**More Frequent Histologic Subtype in Series**	**Preoperative RT**		**Postoperative RT**		**Prognostic Factors in Predicting Worse OS:**	**Factors without Significant Influence on OS:**
		**5 y OS**	**10 y OS**	**5 y OS**	**10 y OS**		
Tanabe et al. [29]	UPS 41%, Liposarcoma 23%, synovial sarcoma 8%	66%	NA			High grade, size > 11 cm, and intraoperative tumor violation	Margins status, Local failure, CT
Wanebo et al. [44]	UPS 20%, Synovial sarcoma 18%, Liposarcoma 17%	59%	NA			High stage, Extent of surgery. For high-grade tumors: size, locoregional extent.	Site, Age, Gender, Histology
Dincbas et al. [46]	Synovial sarcoma 35%, liposarcoma 24% UPS 22%	68.3%	NA			NA	NA
Pao et al. [47]	Liposarcomas + UPS 60%			NA	NA	Stage IV	Margins status, Site, Size, Gender, Age
Talber et al. [24]	Synovial sarcoma 32%, UPS 11%, Epithelioid sarcoma 9%			80%	69%	NA	NA
Le Péchoux et al. [45]	Synovial sarcoma 27%, UPS 21%, Liposarcoma 11%			62% (49–73%)	NA	Size ≥ 5 cm, Margin status	Grade
Khanfir et al. [40]	UPS 30%, Synovial sarcoma 21%			5 y: 77% (69–84)	10 y: 67% (57–76)	High grade (Local and Distant recurrence	Margins status, Use of RT
Issakov et al. [26]	Liposarcomas 100%			NA	67%	NA	Margins status, Site, Age, Gender, Type of liposarcoma
Lee et al. [48]	Liposarcoma 33% Synovial sarcoma 23%, UPS 19%			69.2%	NA	High grade	
McGee et al. [37]	UPS 51%, Liposarcoma 18%			79%	70%	Local control, Age, Gender	Margins status, RT dose (<60/60–66/>66 Gy), Fractionation (hyper vs. conventionnal)
Felderhof et al. [41]	Myxoid liposarcoma 14%, Leiomyosarcoma 13%, Synovial sarcoma 12%			69%	51%	Local recurrence Distant recurrence	Margin status, Site, Size, Grade, Gender, Age, Primary/recurrent, Depth, RT dose (≤56, 60, 66 Gy)
Beane et al. [39]	NA			NA	82% (72–90) 71% (59–81) (20 y)	NA	Grade, Use of RT
Dogan et al. [42]	UPS 26%, Liposarcoma 25%, Synovial sarcoma 13%			71.8%	69.1%	NA	Site, CT, RT dose (<60/≥60 Gy)
	**More Frequent Histologic** **Subtype in Series**	**Preoperative + Postoperative RT**				**Prognostic Factors in Predicting Worse OS:**	**Factors without Significant Influence on OS:**
		**5 y OS**		**10 y OS**			
Schoenfeldet al. [25]	UPS 40%, Synovial sarcoma 17%, Neurofibrosarcoma 9%	96%		91%		NA	NA
Cannon et al. [34]	UPS 42%, Liposarcoma 22%, Synovial sarcoma 13%	NA		62%		NA	NA
Mullen et al. [30]	UPS 22%, Liposarcoma 16%	84% MAID vs. 56% control		66% vs. 38%		No use of neoadjuvant CT	NA
Folkert et al. [35]	UPS 37%, Liposarcoma 28%, Synovial sarcoma 9%	71.7% (66.6–77.2)		NA		NA	IMRT
Kneisl et al. [38]	NA	NA		NA		No use of RT	NA
Roeder et al. [36]	Liposarcoma 31%, UPS 27%, Synovial sarcoma 15%	77%		66%		High grade, metastases at/prior to IOERT	NA
Goertz et al. [27]	UPS 100%	73.0% (64.5–79.7)		NA		High grade Margin Status Depth Age > 60 No adjuvant RT	Size, Gender
Cheng et al. [43]	UPS 45%, Liposarcoma 21%, Synovial sarcoma 12%	Pre 75% ± 15% Post 79% ± 11%		NA	NA	High Stage	Timing of RT, Use of RT

Site corresponds to upper limb vs. lower limbs. Timing of RT corresponds to pre-operative and post-operative RT. Black boxes mean ‘not applicable. CT = chemotherapy; IMRT = intensity-modulated radiation therapy; IOERT = intraoperative electron radiation therapy; MAID = mesna, doxorubicin, ifosfamide, dacarbazine; OS = overall survival; NA = not available; RT = radiotherapy; y = year(s).

### 3.7. Complications

While the standard of care combining limb-sparing surgery and RT, when possible, achieved good LC, complications may occur during and after the treatment. The complications described in all these studies could include wound complications (WCs) (infection, wound dehiscence, hematomas, seromas, etc.), vessel complications such as insufficiency and nerve damage, acute side effects of radiation such as skin reaction and edema, or long-term side effects such as fracture or fibrosis [34,35,38,39,41,42]. These can be the cause of limitations in function and impact quality of life [41]. Nonmanageable complications can lead to amputation. The complication rates of WC, fractures, amputations, and chronic complications are reported in Table 5.

Talbert et al. found a higher complication rate for patients with lesions of the lower extremity than for those with lesions of the upper extremity (28% vs. 56%; *p* = 0.04) [24]. Cannon et al. found a superior incidence of chronic WC for patients with proximal tumors (groin and thigh) than for patients with distal tumors (16% vs. 4%; *p* = 0.008) [34].

The incidence of WC ranged from 8% to 41% for patients treated with preoperative RT, from 2% to 27% for patients treated with postoperative RT and from 2.1 to 5.0% for patients treated with a combination of both techniques. The overall median rate was 18%. For Cheng et al., WC occurred for 75% of patients in the preoperative RT arm and 25% in the postoperative arm (*p* < 0.001) [43]. In the series of Cannon et al., the use of preoperative RT was associated with an acute WC rate of 34% compared to 16% in patients who did not receive RT (*p* < 0.001) [34]. Size > 5 cm was also a significant factor of higher WC in this series (*p* = 0.035) [34]. The presence of WC was associated with a statistically significant decrease in OS in the series of Cheng et al. (*p* = 0.02) [43]. For Alektiar et al., the 5-year WC rate between the RT arm and no RT arm was not significant at 3% vs. 2% (*p* = 0.7) [28], as well as for Beane et al. (*p* = 0.69) [39]. WC grade ≥ 2 did not differ between the IMRT and conventional RT arms in Folkert et al. [35]. Many factors were not significant in predicting WC, such as bone exposure, periosteal stripping, vascular reconstruction, age, and tumor location within the lower extremity [43].

The median rate of fracture incidence was 5.3%; it was 3.3% for patients treated with preoperative RT in the series of Dincbas et al. [46], from 1.1% to 5.3% for those treated with postoperative RT, and from 1.2% to 9% for the combination of both techniques. The median times to fracture ranged from 3.2 years to 7.3 years [31,33,34]. Radiation to the entire circumference of bone was a significant prognostic factor of fracture [33,34]. Surgical exposure of bone, periosteal stripping and anterior compartment location were proposed as significant causes of this complication [33,34]. In contrast, age, tumor site, RT, RT fractionation, dose (≤60 vs. >60 Gy) and RT schedule, beam length (≤30 vs. >30 cm), histology of sarcoma, sex, number of surgeries and use of CT were not predictive of fracture [33,35,37,38]. Dickie et al. specifically analyzed factors linked to fractures. They retrieved as significant patients developing or not developing fracture a higher maximum dose (64 ± 7 Gy versus 59 ± 8 Gy, respectively; *p* = 0.02), mean dose to bone (45 ± 8 Gy versus 37 ± 11 Gy, respectively; *p* = 0.01) and percent of volume of bone receiving ≥40 Gy (V_40Gy_) (76 ± 17% versus 64 ± 22%, respectively; *p* = 0.001). The authors suggested that the risk of fracture may be reduced if the reported factors were constrained below the values of nonfracture patients. IMRT could help to reach these constraints [50]. However, if Folkert et al. did not show any decrease in fracture incidence with IMRT compared to conventional RT, they reported a benefit for dermatitis (48.7% vs. 31.5%; *p* = 0.002) and edema (14.9% vs. 7.9%; *p* = 0.05) [35].

The incidence of amputations ranged from 1% to 25%, with a median rate of 2.6%. Amputations were performed either because of disease progression or because of post-treatment complications. Local relapse was the main indication for amputation in most articles, followed by postoperative complications [24,29,36,44,47]. Incidence of amputations ranged from 1% to 25%, with a median rate of 2.6%. Amputations were performed either because of disease progression or because of post-treatment complications. Local relapse was the main indication for amputation in most articles followed by postoperative complications [24,29,36,44,47].

## 4. Discussion

Previous studies have already demonstrated the benefits of RT for eSTS with well-defined and restricted criteria, leading to improvements in oncological outcomes. The purpose of this systematic review was to use the obtained data of prognostic factors from the selected series to provide compelling evidence supporting the consideration of radiotherapy as a suitable treatment option for a broader range of patients and tumor profiles.

A wide range of prognostic factors was highlighted in this review concerning patient and treatment outcomes. First, as already mentioned in previous prospective randomized trials [11,51] and retrieved in most studies of this review, RT improved LC for patients treated for an eSTS. A dose of at least 60 Gy, with 1.8/2 Gy per fraction, seems necessary to increase LC in the postoperative setting when margins are inadequate, according to Dogan et al. [42]. These data are in accordance with current recommendations [6,7].

Size

If we consider factors that could orient patients to RT, tumors larger than 5 cm seem to benefit from this treatment [32], even though this factor was not systematically retrieved in all series. In contrast, the role of adjuvant RT for patients with tumors ≤ 5 cm in size was not considered positive [52,53]. Perhaps, unlike larger tumors, small tumors already have a good LC rate, which does not allow us to appreciate the benefit of RT.

Margins status

Positive margins were factors that influenced LC, and in some studies, RT decreased its negative impact on LC. Indeed, margin status has been described as one of the most important prognostic factors to have a significant impact on LC in previous studies [53,54,55], also reported in articles of this review [27,29,36]. In the study by McGee et al., patients with close or positive margins did not have a lower LC rate than patients with negative margins. However, it is important to note that patients with close or positive margins received a higher total radiation dose, which could have introduced bias into the results and potentially compensated for the negative impact of positive margin status on LC [37]. Then, positive margins (microscopically and macroscopically) could orient the patient to RT.

Grade

High grade was another significant factor retrieved of lower LC in the studies of Tanabe et al. [29], even if it was not the case for most of the other selected studies in this review [27,32,35,36,40,42,45,47,48].

High grade was also reported as a significant prognostic factor of lower survival and of higher distant relapse in this review and in previous studies [27,29,36,40,44,48,56,57]. Eilber et al. found that a high grade was independently associated with an increased risk of local relapse [57]. Consequently, high-grade tumors are potentially more likely to benefit from RT to reduce the risk of local relapse or even metastasis. The lack of a clear impact of RT on LC and OS in cases of high-grade sarcoma could be hidden by the development of metastasis and premature death compared to the risk of local relapse, explaining why this factor did not appear to be a prognostic factor of LC but did appear to be a prognostic factor of OS.

Depth

Deep tumors were reported as a factor of poor LC [53] and should be irradiated.

Histologic subtype

Khanfir et al. also demonstrated worse LC rates for the UPS histological subtype [40]. Previous studies have shown that UPS is one of the most aggressive subtypes, resulting in a high risk of local recurrence and distant metastasis [18,58,59,60]. Their 5-year OS rates range from 60% to 76% [61,62,63,64]. Similar results were found in this review by Goertz et al., who analyzed only the UPS subtype and demonstrated 5-year LC and OS rates of 58.2% and 73.0%, respectively [27]. Because of its aggressiveness, the UPS histological subtype could benefit from RT. In contrast, liposarcoma was found to be a prognostic factor of better LC than the other subtypes (*p* = 0.004) [65], and among them, myxoid liposarcomas were described to be more radiosensitive than other pathologies [66]. Issakov et al. specifically studied liposarcoma and found excellent results with a 10-year LC of 83% [26]. Some authors, such as Lansu et al., proposed a dose reduction in the preoperative setting from 50 Gy to 36 Gy for myxoid liposarcoma and provided excellent results with a 25-month LC rate of 100% [66].

Given that most of the studies included in this review encompassed a variety of diagnostic subgroups and did not specify tumor characteristics, nor differentiate between local control and survival rates for each subtype, it was not possible to statistically determine the oncological outcomes for each.

Localization

Concerning localization, Alektiar et al. found significantly worse LC rates in patients with proximal tumors (shoulder/groin) compared with noncentral locations (*p* = 0.007) [28]. Perhaps preserving neurovascular bundles for proximal tumors impacts the feasibility of performing a wide local excision compared to distal tumors.

Schedule of RT

The schedule of RT does not seem to impact LC or OS [35,36,43].

Fractionation

Concerning fractionation, LC appeared to be higher with hypofractionation versus conventional fractionation [46]. Greater LC with hypofractionation could be explained by the fact that soft tissue sarcomas have an α/β ratio between 1 and 5 Gy, which implies relative radioresistance to low doses per fraction. However, Koseła-Paterczyk et al. used hypofractionation with preoperative RT and achieved similar LC and complication rates in comparison to conventional fractionation but with a shorter overall treatment time [67].

Finally, prognostic factors of eSTS remain a complex topic involving many variables in widely heterogeneous studies. This heterogeneity included the number of patients in the series, their gender and age, location of the tumor, size, pathology, quality of surgery, and radiotherapy factors; many factors were too numerous to be analyzed in all series. Consequently, logically, the significant impact of a prognostic factor should be considered for the indication of irradiation instead of its non-significance to a non-indication (Figure 2).

Concerning OS, most studies in this review did not find an improvement in survival for the RT arm [35,39,40,45]. These results are in accordance with previous prospective studies that failed to demonstrate an OS benefit for patients treated with RT compared to the other, both for high-grade and low-grade patients [9,11]. We can therefore assume that a higher LC rate due to RT does not necessarily influence OS for several reasons. First, local relapse may not constantly lead to new metastases. Second, local relapse does not necessarily lead to death due to salvage treatments that can be efficiently performed despite their functional consequences, such as amputation [68]. Then, an inconstant impact of LC on OS can be explained by the heterogeneity of many factors. OS may be impacted by local recurrence depending on the location of the primary tumor. Indeed, it has been reported by Gronchi et al. that LC impacts OS in patients with proximal eSTS [69]. The impact of LC on OS was also found in three studies reviewed in this paper [37,40,41]. We can speculate that a locoregional recurrence reaching the thorax or abdomen may cause more lethal damage than a local recurrence distal to a limb, not reaching the viscera. Second, some relapse locations were not accessible to a new efficient and complete surgery or radiotherapy, limiting, consequently, the impact of salvage treatment. Heterogeneity also results in an often non-analyzed difference between R0 and R1/R2 margins concerning metastatic relapse. Indeed, most studies are not designed for this indication. Notably, the rate of metastatic relapse was not higher in patients operated on with R1 margins than in those with R0 margins in the series of Bonvalot et al. [70], and better survival was found with R0 margins [52,71,72]. It remains unclear and inconsistent between studies whether LC is a positive factor of DC and OS. Finally, even if RT does not clearly improve OS, it remains a significant indication in the management of patients with eSTS, as it improves LC.

Chemotherapy

As patients with high-grade extremities are at significant risk for distant recurrence and death from metastatic disease [27,29,36,40,41,43,44,73], CT may be a potential alternative for them. However, the impact of CT on OS or the recurrence of eSTS is controversial, with heterogeneous results [74,75,76]. Most of the studies in this review analyzing the impact of CT on LC [29,30,35,36,40] and on OS [29,42] did not find any significant benefit, whether it was neoadjuvant, concurrent with RT, or adjuvant. This lack of benefit in OS is probably due to the great heterogeneity of the sarcomas included in these trials, with very chemosensitive subtypes such as myxoid liposarcomas and other chemoresistant subtypes such as differentiated liposarcomas [77]. Furthermore, it is possible that this lack of a significant impact on OS could be due to the adverse effects of chemotherapy masking some of its benefits.

In all cases, the decision must be made by multidisciplinary teams at experienced sarcoma centers weighing the risks and benefits.

Complications

WC occurred at least two times more often when using preoperative RT than when using postoperative RT. O’Sullivan et al. demonstrated similar results in a prospective study, with a 35% rate of WC in the preoperative RT arm versus 17% in the postoperative RT arm, with continuity and greater function at 6 weeks after surgery with postoperative RT [12]. However, with a longer follow-up of 2 years, Davis et al. demonstrated that patients treated with postoperative RT had more long-term complications, such as fibrosis, joint stiffness and edema [13]. This could be explained by the increased field size and doses used compared to preoperative RT.

Nevertheless, grading of side effects with the Common Terminology Criteria for Adverse Events (CTCAE) or the Radiation Therapy Oncology Group (RTOG) toxicity scale [78,79] was often poorly used, especially for series published before 2010 [24,34,38,43,44,47,49]. This lack of data led to difficulties in comparing series, mainly those that used modern techniques (IMRT, SIB) or schedules (hypofractionation) of RT. Among the studies that graded side effects, only very few acute and chronic grade 3–5 adverse events were observed [35,37,39,41,42,46]. Adverse events such as postoperative complications or those related to RT have been shown to be mostly manageable with good functional outcomes and should not be an obstacle to the use of RT [36,39,44,47]. Moreover, the impact of all factors that may be the cause of treatment-related adverse events is not always analyzed in detail, such as the type of radiation, the radiation circumference, or the radiotherapy technique, which remained poorly described in these studies. A second limit of this positive observation is the often-short follow-up of the patients, or follow-up chosen for LC but not specifically for late complications that can appear more than five to ten years after irradiation.

IMRT appears to be a good option for treatment due to its better sparing of healthy tissue. Indeed, Stewart et al. demonstrated that using IMRT compared to 3D-CRT allowed significantly lower femur V_45Gy_ and lower normal tissue V_55Gy_ and D_max_ [80]. However, Folkert et al. did not demonstrate a significantly lower rate of fractures and WC but a decrease in radiodermatitis and edema rates [35]. Thus, IMRT appears to be a treatment of choice. This lack of reduction in WC may be explained by the greater difficulty of reducing the RT dose to the skin and subcutaneous tissue without decreasing target coverage for superficial tumors than for deep-seated tumors. A higher WC rate following preoperative RT was also found in the series of Baldini et al. for tumors located less than 3 mm from the skin [81].

It is important to note that most of these studies were conducted retrospectively, leading to inherent heterogeneity, which limits the strength of the conclusions drawn.

Nonetheless, even if it was not possible to establish factor significance, possibly due to a lack of statistical power, the limited adverse events can allow for expanding the indications of RT without being detrimental to the patient when at least one study has demonstrated that a specific factor benefits from RT.

## 5. Conclusions

This review describes the variability of prognostic factors found in the eSTS, which are not consistently retrieved or analyzed among these highly heterogeneous studies. Positive margins, size > 5 cm, deep tumors, and high grades were the most significant factors identified that could benefit from RT, particularly in terms of improving LC. However, for several reasons, achieving LC must be an independent objective when considering radiotherapy for the patient, regardless of DC or OS goals. Additionally, we emphasize the low rates of acute and chronic serious adverse effects of multimodal treatment combining limb-sparing surgery and RT, leading to favorable functional outcomes. Consequently, RT is a relevant treatment option for eSTS, which can also benefit from the advancements in new RT techniques currently available.

## Figures and Tables

**Figure 1 cancers-15-04486-f001:**
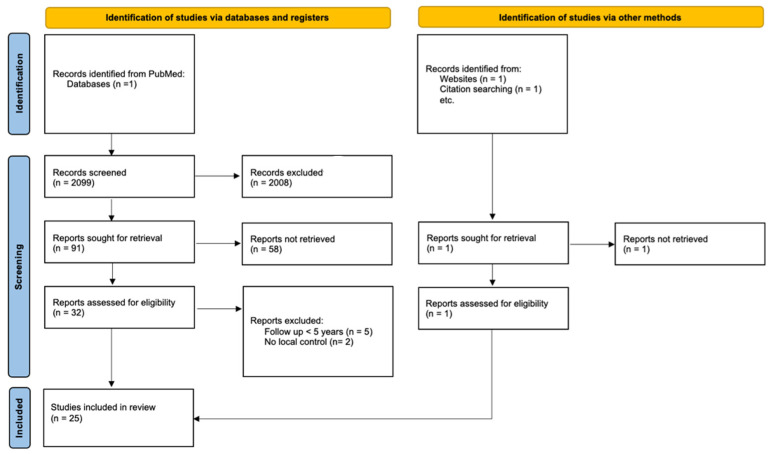
PRISMA flowchart of the literature search and study selection process.

**Figure 2 cancers-15-04486-f002:**
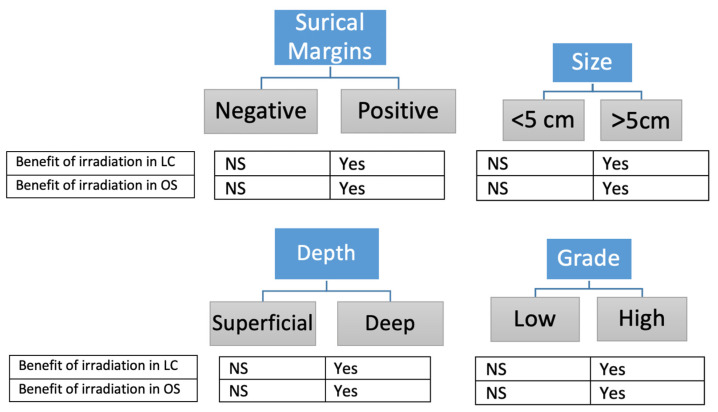
Factors described in the literature (at least one article) in favor of RT. LC = local control, NS = not specifically studied, OS = overall survival.

**Table 2 cancers-15-04486-t002:** Local control and predictive factors of patients with extremity soft tissue sarcomas treated with radiotherapy.

	**More Frequent Histologic Subtype in Series**	**Preoperative RT**		**Postoperative RT**		**Prognostic Factors in Predicting Worse LC:**	**Factors without Significant Influence on LC:**
		**5 y LC**	**10 y LC**	**5 y LC**	**10 y LC**		
Tanabe et al. [29]	UPS 41%, Liposarcoma 23%, synovial sarcoma 8%	83%	NA			High grade tumors, Intraoperative tumor violation, Positive margins	Size, CT, Site
Wanebo et al. [44]	UPS 20%, Synovial sarcoma 18%, Liposarcoma 17%	98.5 (7 y)	NA			NA	NA
Dincbas et al. [46]	Synovial sarcoma 35%, liposarcoma 24% UPS 22%	81%	NA			Conventionnal fractionation vs. hyprofractionation	Margins status, Size, RT
Pao et al. [47]	Liposarcomas + UPS 60%			NA	78%	NA	Margins status, Grade, Stage, Site
Talber et al. [24]	Synovial sarcoma 32%, UPS 11%, Epithelioid sarcoma 9%			80%	74%	NA	Size, Site, Type of RT, RT dose
Le Péchoux et al. [45]	Synovial sarcoma 27%, UPS 21%, Liposarcoma 11%			75% (61–85)	NA	NA	Margins status, Size, Grade
Alektiar et al. [28]	NA			82% (76–88)	NA	Age > 50, Central tumor location (shoulder/groin)	Site, Depth, RT, type of RT (BRT vs. EBRT)
Khanfir et al. [40]	UPS 30%, Synovial sarcoma 21%			78% (70–84)	71% (61–79)	No adjuvant RT, UPS histological type	Margins status, Size, Grade, Depth, Age, Treatment era, Re-excision, CT
Issakov et al. [26]	Liposarcomas 100%			NA	83%	NA	Margins status, Site, Age, Gender, Type of liposarcoma,
Lee et al. [48]	Liposarcoma 33% Synovial sarcoma 23%, UPS 19%			90.7%	NA	NA	Margins status, Grade
McGee et al. [37]	UPS 51%, Liposarcoma 18%			89%	87%	Age > 55 years, Recurrent presentation	Margin status, Stage, Site, RT fractionation, treatment era, RT dose for positive margin
Felderhof et al. [41]	Myxoid liposarcoma 14%, Leiomyosarcoma 13%, Synovial sarcoma 12%			91%	88%	NA	NA
Beane et al. [39]	NA			NA	100% (RT arm)	No RT	NA
Dogan et al. [42]	UPS 26%, Liposarcoma 25%, Synovial sarcoma 13%			77%	70.4%	RT dose, CT	Site, Grade, Stage, Gender
	**More Frequent Histologic** **Subtype in Series**	**Preoperative + Postoperative RT**				**Prognostic Factors in Predicting Worse LC:**	**Factors without Significant Influence on LC:**
		**5 y LC**			**10 y LC**		
Choong et al. [32]	UPS 35%, Liposarcoma 34%, Fibrosarcoma 15%	91.7% (4.4 y)			NA	No RT Size > 5 cm	Margin status, Grade, Depth
Schoenfeldet al. [25]	UPS 40%, Synovial sarcoma 17%, Neurofibrosarcoma 9%	91%			91%	NA	NA
Cannon et al. [34]	UPS 42%, Liposarcoma 22%, Synovial sarcoma 13%	89%			88%	NA	NA
Mullen et al. [30]	UPS 22%, Liposarcoma 16%	NA			MAID:90% (11.2 y) Control:83%	NA	Preoperative CT
Folkert et al. [35]	UPS 37%, Liposarcoma 28%, Synovial sarcoma 9%	92.4% (IMRT) 84.9% (EBRT)				Size > 10 cm, Age > 50 years, IMRT	Margin status, Grade, Depth, Tumor histology, Timing of RT, CT
Roeder et al. [36]	Liposarcoma 31%, UPS 27%, Synovial sarcoma 15%	86%			85%	Positive margins	Site, Grade, Age, Gender, Histology, RT dose, RT timing, primary vs. recurrent, CT
Goertz et al. [27]	UPS 100%	67.6% (RT arm)			NA	No adjuvant RT, Positive margins	Size, Site, Grade, Age, Gender, Depth, Preoperative RT
Cheng et al. [43]	UPS 45%, Liposarcoma 21%, Synovial sarcoma 12%	Pre RT 83% ± 12%		Post RT 91% ± 8%	NA	NA	Timing of RT

Site corresponds to upper limb vs. lower limbs. Timing of RT corresponds to pre-operative and post-operative RT. Black boxes mean ‘not applicable’. BRT = brachytherapy; CT = chemotherapy; EBRT = external beam radiation therapy; IMRT = intensity-modulated radiation therapy; IOERT = intraoperative electron radiation therapy; LC = local control; MAID = mesna, doxorubicin, ifosfamide, dacarbazine; NA = not available; RT = radiotherapy; UPS = Undifferentiated pleomorphic sarcoma; y = year(s).

**Table 5 cancers-15-04486-t005:** Complications of patients with extremity soft tissue sarcomas treated with radiotherapy.

	Timing of RT/Surgery	Wound Complications	Bone Fractures	Amputations	Chronic Complications
Tanabe et al. [29]	Pre	8%	NA	1%	NA
Wanebo et al. [44]	Pre	41%	NA	2.5%	Dysfunction G ≥ 2:4.5% Edema G ≥ 2:7.6%
Dincbas et al. [46]	Pre	20%	3.3%	1.7%	47% Fibrosis: 31.7%, Edema: 13.3% Osteoradionecrosis: 3.3%
Pao et al. [47]	Post	NA	NA	2.5%	Dysfunction G ≥ 2:8%
Talbert et al. [24]	Post	NA	NA	25%	25% Joint stiffness: 5.1% Edema: 1.3%
Le Péchoux et al. [45]	Post	NA	3.2%	0	3-year complication: 55% (41–68) (dysfunction, edema, sclerosis, pain, skin necrosis nerve damage)
Khanfir et al. [40]	Post	NA	NA	0	29% (edema, fibrosis, impairment of joint movement, lymphoedema)
Issakov et al. [26]	Post	NA	5,3%	2.6%	Pain: 68.4% Neuromotor disturbance: 44.7% Joint stiffness: 16.8% Soft-tissue damage: 65.8% Lymphoedema: 21%.
Lee et al. [48]	Post	14%	NA	NA	4.7% (lymphedema and skin ulceration)
McGee et al. [37]	Post	3.4%	6.3%	2.9%	NA
Felderhof et al. [41]	Post	7%	NA	NA	71.1% (all grades) Fibrosis: 55% Joint stiffness: 23%
Beane et al. [39]	Post	27%	2%	6.7%	Dysfunctions G ≥ 2:12% Edema G ≥ 2:25%
Alektiar et al. [28]	Post	2%	NA	0	NA
Dogan et al. [42]	Post	NA	1.1%	NA	Fibrosis: 45.6% Edema: 7.9
Cheng et al. [43]	Pre (43%) + Post (57%)	18% (13% pre, 5% post)	NA	NA	NA
Schoenfeld et al. [25]	Pre (30%) + Post (70%)	NA	4.8%	0	91.3% (edema, fibrosis, joint function)
Cannon et al. [34]	Pre (65%) + Post (35%)	27%	1.2%	NA	10% 20 y radiation-related complication–free survival: 87%
Mullen et al.	Pre (50%) + Pre with postoperative boost (40.1%) + Post (9%)	12.5%	6.5%	2.1%	16.7% (chronic pain, limitations in range of motion, lymphedema)
Folkert et al. [30,35]	Pre (12.2%) + Post (87; 8%)	18.4%	6.9%	NA	Nerve injuries G ≥ 2:2.6% Joint stiffness G ≥ 2:12.9% Edema G ≥ 2:11.3%
Kneisl et al. [38]	Pre (36.9%) + Post (63.1%)	NA	8%	4.9%	NA
Roeder et al. [36]	Pre (17%) + Post (83%)	NA	NA	5%	Dysfunction G ≥ 2:19%
Blaes et al. [33]	Pre (13%) + Post (67%) + both (20%)	NA	9%	NA	NA

Amputations were due to progression of disease or complication of treatment. Timing of RT corresponds to pre-operative and post-operative RT. G = grade; NA = not available; RT = radiotherapy; y = year(s).

## Data Availability

Data available on request due to privacy restrictions. The data presented in this study are available on request from the corresponding author.

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
