# Peer review of "Prognostic Factors in Extremity Soft Tissue Sarcomas Treated with Radiotherapy: Systematic Review of the Literature"

_cancers, 2023, doi:10.3390/cancers15184486_

Round 1
Reviewer 1 Report
The authors present a very comprehensive systemic literature review on the effectiveness of radiation therapy in soft tissue sarcoma treatment.
This is of great value to the field and very well done. I therefore would like to recommend publication of this article.
Author Response
Cf cover letter.
We would like to extend our sincere gratitude to the reviewer for their valuable feedback. We deeply appreciate their positive remarks regarding our systematic review on extremity soft tissue sarcomas. Such encouragement motivates us to continue our contributions to the advancement of knowledge in this field

Reviewer 2 Report
The manuscript made by Lebas A et al is interesting and extensive, they describe several types of eSTS and treatment, focusing mainly in RT therapy.
Authors made and extensive table in which include the 25 studies reviewed in which evaluate treatment and diagnosis made it in which still using MFH instead UPS despite they described previously (patient population) The table 1 is extensive and hard to follow I suggest review the table and try to group the table by diagnosis, for example
Sarcoma
Liposarcoma. (including well-differentiated, dedifferentiated, myxoid, pleomorphic, myxoid-pleomorphic), Age (include, mean, min - max and standard deviation), sex (with preference relation between M:F, for example 2:1 (M50 - F25), median follow including average, tumor size (only one measure, related to average for example: >8 cm or <8cm, Most frequent RT used in the liposarcomas, according to articles reviewed.
On the table 2 and 3 include if it posible the diagnosis (synovial sarcoma, fibrosarcoma, etc...)
The discussion need be better explained, resalting better the results obtained, for example the predominant surgical margins were R2, in which the leimyosarcoma and fibrosarcoma were the predominant sarcomas with this margins, the RT for this type of tumors with this margins was _________ and the response to treatment.
Overall the manuscript it is interesting, but it is extensive and hard to follow, the discussion need be more focus on the results obtained in their review, maybe include graphics in which highlight the results obtained in their review.
English has minor issues that need to be reviewed.
Author Response
Cf cover letter.
The manuscript made by Lebas A et al is interesting and extensive, they describe several types of eSTS and treatment, focusing mainly in RT therapy.
1) Authors made and extensive table in which include the 25 studies reviewed in which evaluate treatment and diagnosis made it in which still using MFH instead UPS despite they described previously (patient population).
Reply:
We are greatly thankful to the reviewer for their valuable advice. In accordance with their guidance, we have made the change from 'MFH' to 'UPS' throughout the entire manuscript.
2) The table 1 is extensive and hard to follow I suggest review the table and try to group the table by diagnosis, for example
Sarcoma
Liposarcoma. (including well-differentiated, dedifferentiated, myxoid, pleomorphic, myxoid-pleomorphic), Age (include, mean, min - max and standard deviation), sex (with preference relation between M:F, for example 2:1 (M50 - F25), median follow including average, tumor size (only one measure, related to average for example: >8 cm or <8cm, Most frequent RT used in the liposarcomas, according to articles reviewed.
Reply:
We have made several corrections to Table 1. Specifically, we merged the 'authors' and 'date of publication' columns into a single 'series' column. In addition, we reordered the studies based on their sample size, arranging them from the highest to the lowest. Furthermore, we have homogenized the 'histologic subtype' column in the table, retaining only the most common subtypes.
It's important to note that due to the numerous diagnoses present in each study and the absence of detailed tumor and patient characteristics categorized by diagnosis, we found it impractical to classify the data by pathology. We believe these changes to the table enhance its quality and hope they meet with your approval
3) On the table 2 and 3 include if it possible the diagnosis (synovial sarcoma, fibrosarcoma, etc...)
Reply:
We have incorporated a diagnosis column into tables 2 and 3 as suggested by the reviewer. We would like to extend our appreciation to the reviewer for this valuable comment, which not only enhances the quality of the manuscript but also makes it more reader-friendly.
4) The discussion need be better explained, resalting better the results obtained, for example the predominant surgical margins were R2, in which the leimyosarcoma and fibrosarcoma were the predominant sarcomas with these margins, the RT for this type of tumors with these margins was _________ and the response to treatment.
Overall, the manuscript it is interesting, but it is extensive and hard to follow, the discussion need be more focus on the results obtained in their review, maybe include graphics in which highlight the results obtained in their review.
Reply:
We have restructured the discussion section to improve its readability, with a particular focus on factors that can guide patients toward radiotherapy. Additionally, we have included a figure to complement the discussion text. We trust that these extensive revisions will meet the reviewer's expectations and are hopeful for their approval.
5) English has minor issues that need to be reviewed.
Reply
Before the first submission, the manuscript has undergone a comprehensive revision by experts from AJ. After the comment of the reviewer, we included again a thorough proofreading process. We have taken these steps to ensure the highest quality of our manuscript.

Reviewer 3 Report
The authors conducted a systematic review to clarify the importance of the prognostic factors in a subset of patients treated with eSTS, and to provide valuable insight to optimize radiotherapy treatments.
The authors described several factors and significant factors that could orient patients to radiotherapy were highlighted. These positive prognostic factors provided valuable insight to optimize radiotherapy treatments for patients treated for soft tissue sarcoma of the extremities.
The article fills a gap in the current literature.
Materials and methods section is well written.
Results section is very relevant and well detailed.
Discussion section is very well written, and the authors well deepened all the topics.
Overall considered it is a good article and I accept it in the current format.
Author Response
Cf cover letter.
We extend our heartfelt gratitude to the reviewer for their insightful comment. As highlighted, we firmly believe that this article offers valuable insights into the optimal management of extremity soft tissue sarcomas and contributes to a better understanding of their heterogeneity. We aspire that this review will serve as a valuable resource to assist clinicians in guiding patients toward radiotherapy.

Round 2
Reviewer 2 Report
This second revision of the manuscript made by Lebas A. et al. was improved over the first version. The authors took into account all of my suggestions. The manuscript is now better understandable, the tables have been improved a lot, and it is now publicable in the cancers journal. The authors showed an English certification (AJE) that ensures the quality of the language.